# Forests Attenuate Temperature and Air Pollution Discomfort in Montane Tourist Areas

**Elena Gottardini** [1,*], **Fabiana Cristofolini** [1], **Antonella Cristofori** [1] **and Marco Ferretti** [2]

1   Research and Innovation Centre, Fondazione Edmund Mach (FEM), 38098 San Michele all'Adige, Italy
2   Swiss Federal Institute for Forest, Snow and Landscape Research WSL, Zürcherstrasse 111,
    CH-8903 Birmensdorf, Switzerland
*   Correspondence: elena.gottardini@fmach.it

**Abstract:** Forests deliver many ecosystem services, from provisioning to regulating and cultural services. We aimed at demonstrating microclimatic regulation and pollutant removal as especially relevant ecosystem services when considering the tourism vocation of the Alpine regions. A study was realized along an altitudinal gradient (900–1600 m a.s.l.) in Trentino, northern Italy, an area with high touristic presence (ca. 9.3 million overnight stays in summer 2021). Nitrogen dioxide ($NO_2$, $\mu g\ m^{-3}$), ozone ($O_3$, $\mu g\ m^{-3}$) concentrations, air temperature (T, °C), and relative humidity (RH, %) were simultaneously measured in three open-field sites (OF) and below-canopy Norway spruce forest stands (FO) during the period 23 May–7 August 2013. The temperature–humidity index (THI) was calculated. We found a distinct mitigating effect of forest on T, with lower maximum (−30.6%) and higher minimum values (+6.3%) in FO than in OF. THI supported a higher comfort sensation in FO than in OF, especially in the central part of the day. $NO_2$ concentrations did not differ between OF and FO; ozone concentrations were lower in FO than OF. This study confirms the role of forests in providing several ecosystem services beneficial for forest users, especially relevant for promoting nature-based tourism in the Alpine region.

**Keywords:** forest; ecosystem service; temperature mitigation; air quality; human wellbeing

## 1. Introduction

Forests deliver provisioning, cultural, and regulating ecosystem services, resulting in several related benefits [1]. Provisioning services include food (plants, animals), fuel (wood), biochemicals, natural medicines, and pharmaceuticals. Cultural ecosystem services refer to non-material benefits and comprise, among others, the aesthetic value, the recreation, and ecotourism relevant to the human wellbeing and health [2]. The positive effect of forests on physical, mental, and social health is well-known (see e.g., https://efi.int/forestquestions/society (accessed on 8 March 2023)), and the willingness of forest users to pay for enjoying such benefits [3,4] underlines the great value of this ecosystem. Regarding the regulating services, besides air quality control and protection from flooding and erosion [5–7], forests influence the micro- and macro-climate at the global (carbon sequestration) and local (climate regulation on temperatures and precipitation) scale [8–10].

Biophysical properties of forests promote local climate stability by reducing extreme temperatures in all seasons and times of day [11]. Based on models, Gohr et al. [8] suggest that replacing 10% of agricultural land with forest would reduce the mean temperature by 0.9 °C. While the effectiveness of vegetation in reducing air temperature is well-known and documented for urban green spaces [12–14], less evidence is available for remote forest sites [15].

Forests may have an impact on human thermal sensation, which is influenced by the combination of microclimate variables such as solar radiation, wind speed, and relative humidity, as well as subjective perception [13]. The temperature–humidity index (THI) is largely used in urban and farm systems [13,16–18] for evaluating the only physiological

perspective of comfort on the basis on easily available data such as air temperature (T) and relative humidity (RH). Recently, THI has been used to evaluate climate suitability, which concurs, together with the landscape aesthetic quality and recreation utilization, to assess the recreation services offered by the Tibetan Plateau [19].

Forests may play an important role in ameliorating air quality through pollutant removal. Air pollutants negatively impact on human health and wellbeing. Exposure to nitrogen dioxide ($NO_2$) and tropospheric ozone ($O_3$) is strongly associated with respiratory symptoms, eye and nose irritation, decrease in immune defense, and other pathological effects, including an increase in mortality due to respiratory and cardiovascular diseases [20,21]. Both $NO_2$ and $O_3$ are also phytotoxic gaseous pollutants, causing damage to plants [22,23]. $NO_2$ is a primary pollutant resulting from the combustion of fossil fuels, and, thus, is expected at high concentrations in dense urban areas and their surroundings, and along high traffic roads, and not in remote areas. In addition, $NO_2$ is a precursor of ground-level particulate matter, acid rain, and ozone; the latter can be transported far from the emission source to remote areas, including mountainous and forest areas.

Although the EU clean air policy (https://ec.europa.eu/environment/air/index_en.htm (accessed on 8 March 2023)) aims to reduce exposure to air pollution by setting objectives for minimizing the harmful effects of air pollution on health and environment, the quality of life of the European population is still at risk because air quality standards are not being met [24–26]. For example, in Europe in 2013, 16,000 premature deaths were estimated to be attributable to ozone [27]. Forest vegetation can act as a sink for pollutants, in particular for ozone through stomatal and non-stomatal processes [28].

In these respects, robust qualitative and quantitative data on the benefits offered by forests are of great importance: (i) for information and awareness campaigns about the benefits that forest users can enjoy in attending this environment; (ii) to support studies about specific forest services, beneficial to human health; (iii) to study the effects in order to select and adopt a sustainable forest management capable of maximizing the concerned services; and (iv) for policy makers and destination managers to promote forest areas for tourism purposes.

In this study, the hypothesis to be tested is that in Norway-spruce-dominated montane forest, stands air temperature and air pollution are significantly lower than in neighboring open-field sites. To this purpose, in situ measurements were carried out in forested sites and adjacent not-wooded open fields, considering nitrogen dioxide, tropospheric ozone, air temperature, relative humidity, and the derived temperature–humidity index (THI). Renaud and Rebetez [15] previously adopted a similar study design (i.e., open-site vs. below-canopy), but were limited to meteorological measurements. The application of THI in forest, up to now poorly used in a similar environment [29], could represent—along with air pollutant removal quantification—a novelty to characterize the forest ecosystem services and valorize, e.g., ecotourism destinations.

This study will provide new insights on cultural and recreational aspects of forests, beneficial to human wellbeing and health. This is especially relevant for a highly touristic alpine area as the Alps, a holiday destination for around 120 million guests each year (www.alpconv.og (accessed on 1 July 2022)), attracted—among other characteristics—by climate and clean air.

## 2. Materials and Methods

### 2.1. Study Area and Design

The study was carried out in Trentino, northern Italy, a 6.2 km$^2$ alpine region, climatically characterized by a combination of fully humid–warm temperate, snow, and frost climate as the altitude increases from the valley bottoms (min 65 m a.s.l.) to the high mountains (max 3769 m a.s.l.) [30]. A total of 55% of the territory is covered by forest, mainly (67%) represented by conifers, with Norway Spruce (*Picea abies* Karst), Silver fir (*Abies alba* Mill.), Larch (*Larix decidua* Mill.), Austrian Pine (*Pinus nigra* Arnold), and

Scots pine (*Pinus sylvestris* L.) as dominant species; the remaining 33% is represented by broad-leaved species [31].

The region is highly attractive for tourists, with 9.3 million overnight stays registered during June–September 2021 (http://www.statistica.provincia.tn.it/statistiche/settori_economici/turismo/ accessed on 13 January 2022). The study design was set on three randomly selected open-field sites (OF) along a E–S exposed forested slope within three altitudinal ranges, from 800 to 1600 m a.s.l.; for each OF, three surrounding forest sites (FO) within a radius of min 60 m and max 260 m were randomly selected (Figure 1; Table 1; for more details on the study design see Gottardini et al. [32]). The altitudinal range considered represents the optimal distribution range of Norway spruce [33], the tree species that dominates the forest stands where the study was carried out.

### 2.2. Field Measurements

In each OF ($n = 3$) and FO point ($n = 9$), microclimatic measurements (air temperature and relative humidity) were performed by means of data loggers (Tinytag Ultra, Gemini Data Loggers Ltd., Chichester, UK), installed at 1.5 m above ground level, and programmed to record T (°C) and RH (%) every 15 min over the entire study period, from 23 May to 7 August 2013. Nitrogen dioxide ($NO_2$) and ozone ($O_3$) concentrations were measured on a weekly basis over an 11-week period by means of passive samplers (http://www.passam.ch 1 February 2023). The $NO_2$ and O3 samplers consist of polypropylene tubes. $NO_2$ is collected by molecular diffusion along an inert tube to an absorbent (triethanolamine) and determined spectrophotometrically by the Saltzman method [34]. The diffusive sampler for $O_3$ is a polypropylene tube containing a glass fiber filter, dipped in a solution of 1,2-di(4-pyridyl)-ethylene (DPE) in acetic acid. DPE reacts with $O_3$, yielding an aldehyde, the amount of which is finally determined spectrophotometrically. Both $NO_2$ and $O_3$ samplers were placed in a special shelter to protect them from rain and minimize the wind influence, and concurrently exposed with meteorological data loggers.

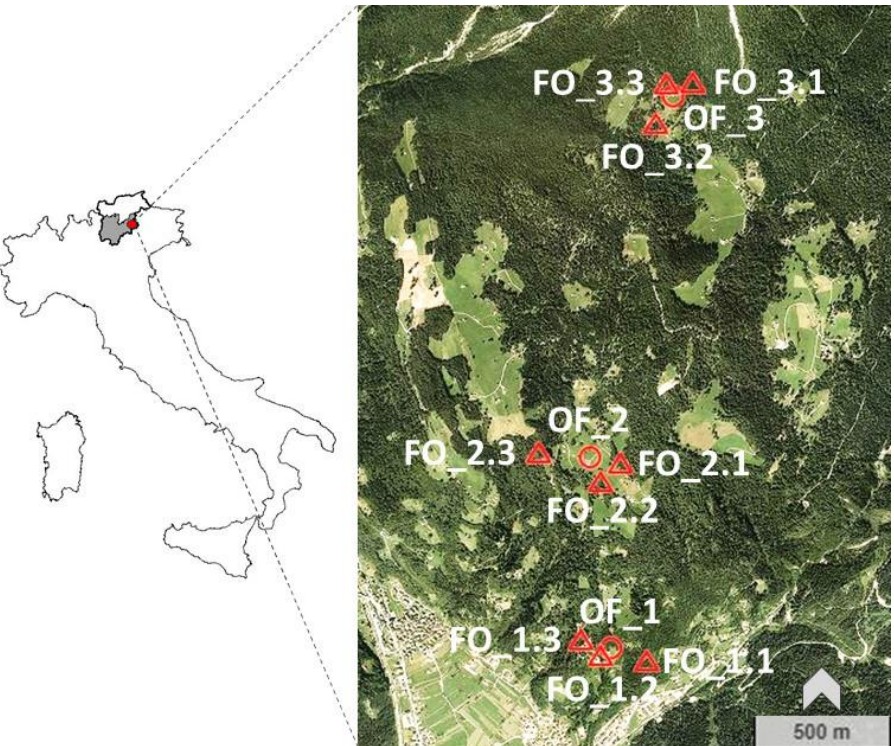

**Figure 1.** Location of the study site in Italy and spatial arrangement of the measurement points in forest stands (FO) and open field (OF) along the elevation range.

**Table 1.** Geographical position of the measurement points located in the open field (OF) and in three surrounding forest stands (FO) in each of the three elevation ranges; for the measurement points in FO, the distance from the OF is reported.

| Elevation Range, m a.s.l. | Measurement Point | Elevation, m a.s.l. | Long. | Lat. | Distance from OF, m |
|---|---|---|---|---|---|
| | OF_1 | 913 | 11.84407 | 46.18728 | - |
| 800–900 | FO_1.1 | 907 | 11.84635 | 46.18671 | 188 |
| | FO_1.2 | 908 | 11.84332 | 46.18703 | 64 |
| | FO_1.3 | 927 | 11.84213 | 46.18779 | 160 |
| | OF_2 | 1177 | 11.84307 | 46.19578 | - |
| 1100–1200 | FO_2.1 | 1210 | 11.8451 | 46.19569 | 158 |
| | FO_2.2 | 1179 | 11.84353 | 46.1949 | 104 |
| | FO_2.3 | 1185 | 11.8398 | 46.19627 | 258 |
| | OF_3 | 1521 | 11.84935 | 46.21214 | - |
| 1500–1600 | FO_3.1 | 1544 | 11.85066 | 46.21268 | 118 |
| | FO_3.2 | 1504 | 11.84818 | 46.21095 | 159 |
| | FO_3.3 | 1547 | 11.84896 | 46.2127 | 71 |

*2.3. Data Analysis*

Data were checked for plausibility; RH values equal to zero were discharged because of implausibility. THI was calculated according to Nieuwolt [35]:

$$\text{THI} = (0.8 \times \text{T}) + \left( \frac{\text{T} \times \text{RH}}{500} \right) [°\text{C}]$$

THI values were referred to from the thermal sensation scale proposed by Deosthali [36].

To test for possible significant differences between OF and FO, the *T*-test for dependent samples was used for daily T, RH, and THI, and the Wilcoxon matched pairs test was used for weekly $NO_2$ and $O_3$. One-way ANOVA (for daily values) or Kruskal–Wallis (for weekly values) analysis was applied to investigate whether differences occurred among the elevation ranges, and among the three FO measurements for each elevation range. In case of significant differences, the multiple comparison was applied. Data from the three FO points of each altitudinal range were averaged, after verifying no significant differences ($p > 0.05$) existed. Statistical analyses were carried out using Statistica software (TIBCO Statistica® 13.3).

**3. Results**

Microclimatic measurements were analyzed in relation to elevation and tree cover (OF vs. FO). Besides the superimposed elevation gradient of $-0.4$ to $-0.8$ °C per every 100 m elevation gain, common in OF and FO, we observed that T mean and T max were higher in OF than in FO in all the three elevation ranges (Table 2).

Averaged over the three elevation ranges, T mean and T max in the OF are significantly higher than in FO ($p < 0.001$), with differences of 2.4 °C for T mean and 8.5 °C for T max, while minimum temperature is significantly ($p < 0.001$) lower in OF than in FO ($-6.3\%$). The daily min–max T range in FO is always narrower in comparison to the OF, especially marked for T max (Figure 2a).

**Table 2.** Mean, min, and max daily air temperature (T), relative humidity (RH), and temperature–humidity index (THI), and mean weekly nitrogen dioxide ($NO_2$) and ozone concentration ($O_3$) in the open field (OF) and in the surrounding forest points (FO; mean of the three points for each OF) averaged over the entire study period (23 May–7 August 2013) for the three sites located in the different elevation ranges. The percentage differences between the averaged values for the three OF and the related FO points (Mean ΔOF-FO), and the significance of the differences ($t$-test for T, RH, and THI; Wilcoxon test for $NO_2$ and $O_3$; n.s. $p > 0.05$; ** $p < 0.01$; *** $p < 0.001$) are also reported.

| | Elevation Range, m a.s.l. | | | | | | | | | |
|---|---|---|---|---|---|---|---|---|---|---|
| | 800–900 | | | 1100–1200 | | | 1500–1600 | | | Mean ΔOF-FO |
| | OF (n = 1) | FO (n = 3) | ΔOF-FO | OF (n = 1) | FO (n = 3) | ΔOF-FO | OF (n = 1) | FO (n = 3) | ΔOF-FO | |
| Mean T, °C (n = 77) | 17.9 | 15.7 ± 0.53 | 2.3 | 16.4 | 14.0 ± 0.38 | 2.3 | 14.6 | 12.1 ± 0.38 | 2.5 | +14.5% *** |
| Min T, °C (n = 77) | 11.0 | 11.3 ± 0.10 | −0.3 | 9.0 | 10.2 ± 0.06 | −1.2 | 8.3 | 8.6 ± 0.07 | −0.3 | −6.3% *** |
| Max T, °C (n = 77) | 29.6 | 21.6 ± 1.52 | 8.0 | 29.5 | 19.5 ± 1.01 | 10.0 | 24.2 | 16.7 ± 1.25 | 7.5 | +30.6% *** |
| Mean RH, % (n = 77) | 72.6 | 82.0 ± 2.84 | −9.4 | 77.6 | 84.2 ± 1.99 | −6.6 | 78.5 | 85.4 ± 2.10 | −6.9 | −10.0% *** |
| Min RH, % (n = 77) | 44.4 | 56.6 ± 0.73 | −12.2 | 40.7 | 59.9 ± 0.22 | −19.2 | 49.4 | 64.2 ± 0.31 | −14.8 | −34.3% *** |
| Max RH, % (n = 77) | 97.9 | 96.6 ± 7.23 | 1.3 | 99.7 | 97.5 ± 6.98 | 2.2 | 99.0 | 97.4 ± 6.87 | 1.6 | +1.7% *** |
| Mean *THI*, °C (n = 77) | 18.0 | 15.0 ± 0.42 | 3.0 | 15.3 | 13.5 ± 0.33 | 1.9 | 14.1 | 11.7 ± 0.33 | 2.5 | +15.5% *** |
| Min *THI*, °C (n = 77) | 11.8 | 11.1 ± 0.09 | 0.6 | 9.0 | 10.1 ± 0.06 | −1.1 | 8.4 | 8.5 ± 0.07 | −0.1 | −2.1% ** |
| Max *THI*, °C (n = 77) | 26.3 | 19.7 ± 1.18 | 6.6 | 26.0 | 17.9 ± 0.83 | 8.1 | 21.8 | 15.4 ± 1.06 | 6.4 | +28.5% *** |
| Mean $NO_2$, µg m$^{-3}$ (n = 11) | 2.2 | 2.2 ± 0.42 | 0.0 | 1.1 | 1.2 ± 0.37 | −0.1 | 1.1 | 0.9 ± 0.15 | 0.2 | +1.7% n.s. |
| Mean $O_3$, µg m$^{-3}$ (n = 11) | 71.1 | 59.0 ± 3.16 | 12.1 | 62.5 | 68.9 ± 3.18 | −6.3 | 80.5 | 71.9 ± 5.51 | 8.6 | +6.7% ** |

When T max is above 30 °C (up to 40 °C) in OF, it is between 20 and 30 °C in FO; on the other side, T min is almost the same in OF and FO (Figure 3).

The T mean mitigation provided by forest ($\Delta T_{mean}$, i.e., the difference between daily mean air temperature measured in the OF and in FO) is similar among the three elevation ranges ($p = 0.082$). However, between 8 a.m. and 10 p.m., T mean is consistently higher in OF (on average, 20.1 °C) than in FO (15.7 °C), with a T abatement in FO ranging between 4.8% and 30.7% (Figure 4a). On the contrary, during the night (from 11 p.m. to 7 a.m.) T mean is slightly higher in forests (on average, 11.5 °C) than in open field (on average, 11.0 °C).

Throughout the entire study period, daily mean and minimum relative humidity (RH) are, on average, higher in FO than in OF (Table 2) (Figure 2b). On the contrary, the RH max results in OF are significantly higher than in FO. From 7 am to 11 pm, RH mean results (Figure 4b) are significantly higher in FO than in OF (77.6% vs. 65.7%), a difference that is not observed overnight.

The THI mean values of the three sites are represented in relation to the thermal comfort sensations during the study period (Figure 2c). Up to the beginning of June, mean THI values, both in FO and OF, lay in the cool–slightly cool sensation range. In the same period, THI max in OF peaks in the slightly warm range. Later in June, THI increases, and maximum values reach the warm sensation. During July and early August, THI max in OF is almost in the slightly warm sensation range, with frequent peaks in the warm belt. The daily THI mean pattern was calculated for two distinct periods, 23rd May to 30th June (i.e., late spring) and 1 July to 7 August (i.e., midsummer) and compared with the reference ranges defined for human sensation (Figure 5).

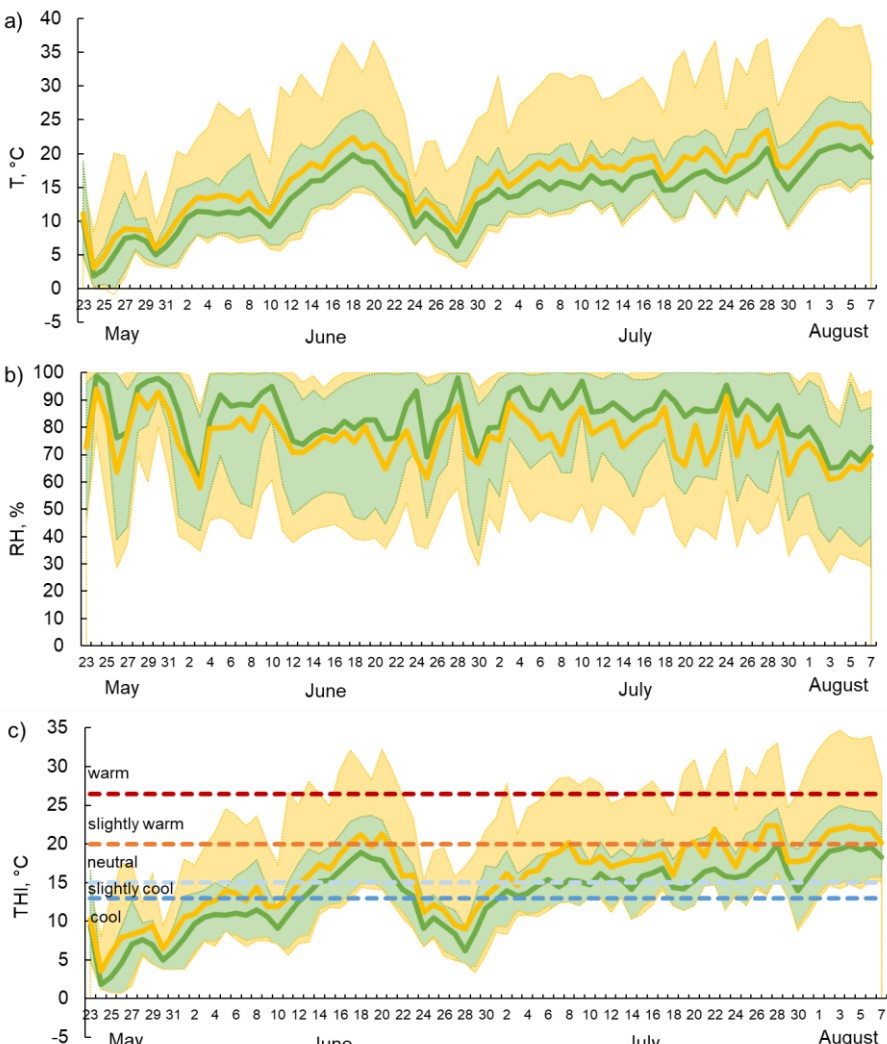

**Figure 2.** Mean daily air temperature (T; (**a**)), relative humidity (RH; (**b**)) and temperature–humidity index (THI; (**c**)) averaged for the three sites. Solid lines represent the mean values registered in open field (orange) and in forest (green); light orange and green areas represent the minimum and maximum temperature in open field and in forest, respectively. Dashed lines in (**c**) delimit the ranges of thermal comfort sensations as defined by Deosthali (1999; in: Ruiz and Correra, 2015).

During the late spring (Figure 5a), the more comfortable hours in FO are between 1 p.m. and 6 p.m., when the perceived temperature is slightly cool (13.7 °C); the best feeling in OF appears to be between 10 a.m. and 6 p.m., when THI mean is neutral (17.2 °C). During the summer period, 1 July–7 August (Figure 5b), THI mean in FO is neutral between 10 a.m. and 12 p.m. (17.9 °C), while in the OF, THI is slightly warm between 10 a.m. and 7 p.m. (23.5 °C), and neutral–slightly cool during the rest of the day.

As for air pollutants, nitrogen dioxide was found in very low concentrations in both FO and OF in all the three sites, with no significant differences between these two environments during the study period (Table 2). Nitrogen dioxide values range from 2.2 μg m$^{-3}$ in the lowest elevation range, both in FO and OF, to 0.9 μg m$^{-3}$ in FO and 1.1 μg m$^{-3}$ in OF in the highest elevation range. On the contrary, ozone concentration is, on average, significantly ($p < 0.05$) higher in OF than in FO (Table 2), and this pattern is evident during the entire study period, with differences ranging between 2% and 13%. As expected, ozone concentrations increase with altitude, both in FO and OF (+22% and +13%, respectively, from the lowest to the highest elevation range).

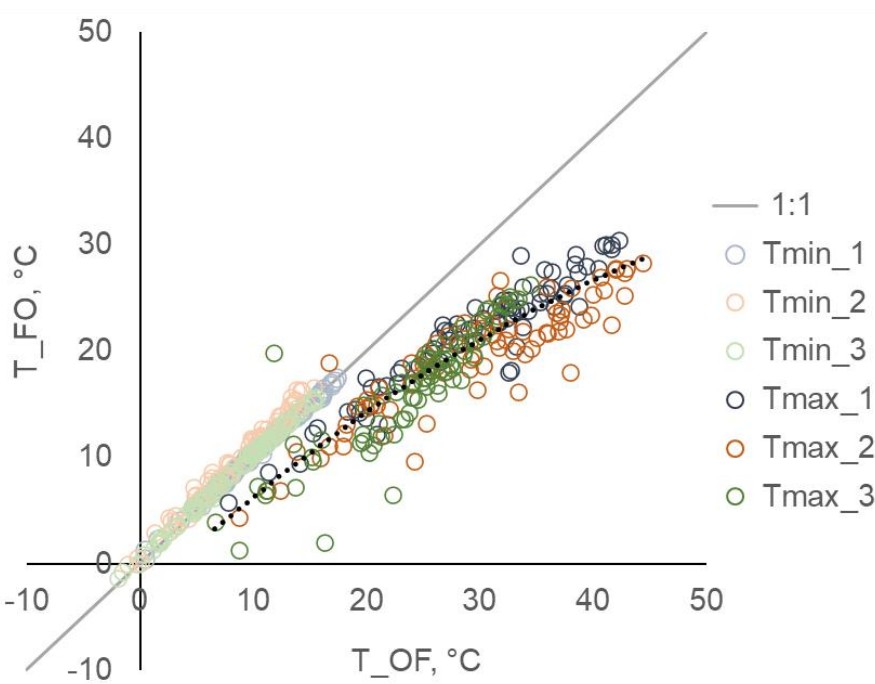

**Figure 3.** Minimum and maximum daily air temperature registered in the three open-field sites (OF; x axis) and in the correspondent forest stands (FO; y axis; mean of the three forest measurement points).

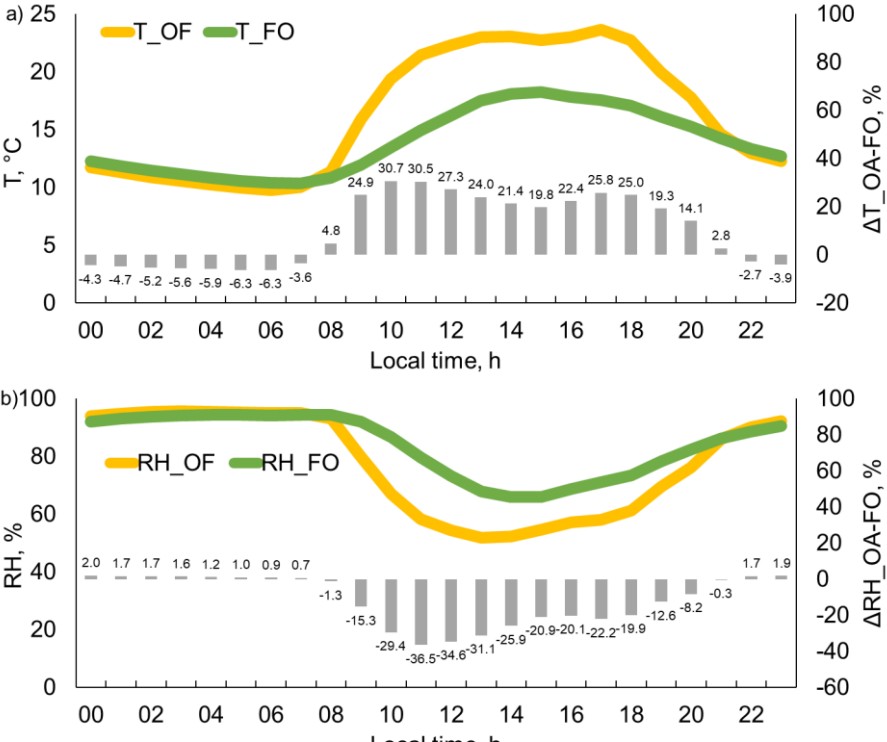

**Figure 4.** Hourly mean values of (**a**) air temperatures and (**b**) relative humidity measured in the open fields (OF; yellow line) and in the surrounding forest (FO; green line) (mean of the three altitudinal ranges over the entire study period, 23 May–7 August 2013).

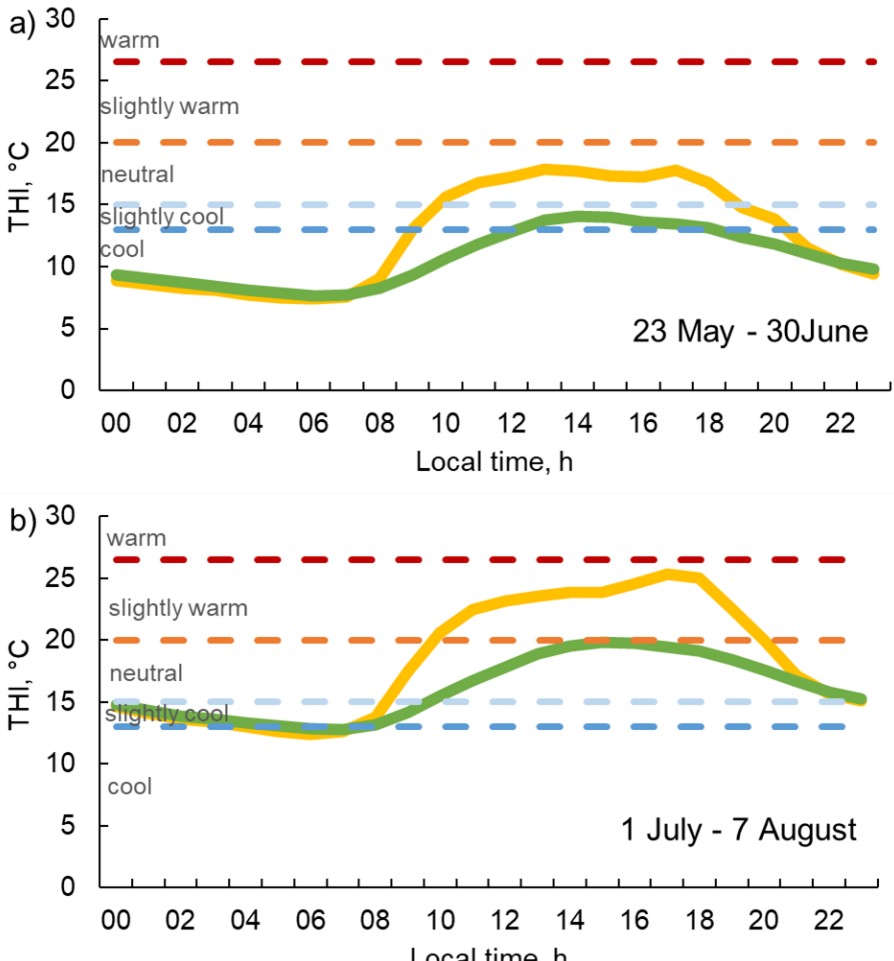

**Figure 5.** Hourly mean of THI values calculated for the open fields (OF; yellow line) and the surrounding forest sites (FO; green line) (mean of the three sites) as average of the three sites for (**a**) the 23 May–30 June (i.e., late spring) and (**b**) 1 July–7 August (i.e., early–midsummer) periods; dashed lines delimit the ranges of thermal comfort sensations as defined by Deosthali (1999; in: Ruiz and Correra, 2015).

## 4. Discussion

In this study, a distinct effect of Norway-spruce-dominated forest on air temperature mitigation was proved, especially marked for T mean (−2.4 °C) and T max (−8.5 °C), with the latter much more substantial in comparison to what was previously reported by Renaud and Rebetz (2009) for the spruce forest type (ΔT max: −1.29 and −2.47 K). When maximum daily temperature in the open fields is higher than 30 °C (up to 40 °C), the corresponding values in forest remain approximatively between 20 °C and 30 °C, which is considered an optimal range for tourism purposes, as reported by Amelung and Viner [37]. The role of forest in temperature mitigation assumes a great relevance especially in the context of climate change. The increase in frequency of heat waves [38] and warm air masses [39] observed in Europe in the last four decades underlines the urgent importance to implement mitigation and adaptation actions. The cooling effect of forests may also have an important economic value, quantifiable by, e.g., combining simple meteorological data and regional electrical cooling costs [40]; such economic value represents an exploitable asset, for instance to further valorize forest ecosystem services and promote forested territories with a particular touristic vocation, enhancing their attractiveness as summer destination.

The ongoing climate crisis will increase people's outdoor heath stress, discouraging outside activities [41,42]. A gradual increase in both the annual and seasonal temperature–humidity index is expected up to 2050 in the Mediterranean basin, particularly marked

for Spain, southern France, and Italy [43]. In this study, THI supported a higher comfort sensation in forest stands than in open fields during the assessed period and in the central part of the day. Due to climate change, summer tourism in the Alps is expected to increase in the future thanks to the more comfortable range of temperatures compared to lowlands [44]. Beside the thermal comfort offered by forests during warm summer days, a concurrent reduction in $O_3$ concentrations was documented. Since ozone concentrations generally increase with the altitude [45], ozone removal by forests at the higher elevations assumes even greater importance.

The documented negligible presence of $NO_2$, declining with the altitude, together with $O_3$ removal and temperature mitigation, represents one of the most valuable forest ecosystem services relevant for human wellbeing and health. To the best of our knowledge, this study is one of the first attempts at concurrently quantifying temperature and air pollution discomfort attenuation by on-site measurements in Alpine forests.

Overall, the ecosystem services provided by forests can contribute to attenuating the negative effects of global change and improve the wellbeing of people. This is especially relevant in the tourism and recreation sectors, since climate is an important factor in choosing destination and type of vacation and travel experiences [46], and more in general for forest users attending forests for different purposes. The increasing attention for a healthy and sustainable lifestyle includes free time, with the choice of destinations and nature-based vacations capable of providing positive wellbeing experiences. In this respect, Trentino region—with 373,259 ha of forests (Italian Forest Inventory 2015)—and, more generally, the Alpine area, of which more than 40% is covered by forests, is selected by tourists for the valuable environmental heritage and the numerous opportunities for outdoor activities. In the future, the mountain tourism industry will likely have to face a longer warm season and a higher frequency of hot days, and so could take the opportunity to promote initiatives and activities to be carried out in the forest, exploiting the quantifiable thermal and air quality benefits as strengths. Forest therapy, considered to play an important role in human health promotion and disease prevention [47], is a good example of exploitation of forest ecosystem services.

Despite the limitations in time and space coverage, this study highlights, under specific circumstances, the relevance of punctual and measured microclimate effects of forests, valuable as an example to encourage ecotourism in forest areas during the main period of summer heat. Considering climate change and the human search for advantageous conditions to wellbeing, such an approach could be exploited on a wider spatial and temporal scale, and for different purposes. For instance, it could be of interest to investigate if species composition or different management systems (e.g., pro-forestation [48]) have an impact on temperature and air pollution mitigation of forests. The outcomes could be relevant in planning specific forest areas (e.g., for outdoor learning and education, natural connections, forest bathing, artistic initiatives), or to design differentiated outdoor parkours and activities (e.g., walking, hiking, mountain biking, orienteering, etc.) according to the season and the time of the day.

This approach would represent a useful novelty and tool for local tourism authorities and forest managers to promote sustainable ecotourism.

## 5. Conclusions

Several studies deal with the capacity of plants to lower air temperatures during the summer by absorbing a fraction of incoming sunlight and by transpiration, and with the theoretic mechanisms of air pollutant removal by plants through both stomatal and non-stomatal processes.

This study, based on in-field measured data, aimed to demonstrate the actual microclimate regulation and pollutant removal provided by forests. Focusing on Norway-spruce-dominated forests, the results confirm the effective reduction in air pollution (−7% for ozone) and air temperature (−15% on average, and −30% for maximum values) in forest stands in comparison to open fields. Moreover, the thermal comfort perceivable in an alpine

forest during the warmest hours in summer is underlined by the THI values calculated for the study area.

Regulating air quality and temperature, which alleviate the discomfort of local population and tourists, are two of the most relevant forest ecosystem services impacting on human health and mental wellbeing, also in the light of the global trend of rising air temperature and heat wave frequency, together with the persistence of high levels of ozone.

A documented cleaner air, combined with a thermal comfort in summer, as demonstrated by our study, makes the forested environment a place capable of giving actual psycho-physical benefits to people. This evidence has clear practical implications, especially relevant for the Alpine region that may become even more attractive in summer due to the expected rapid increase in air temperature. In general, considering the priorities identified by the EU tourism policies, the role of forests in providing regulating and cultural ecosystem services could be better promoted to favor sustainable, high-quality destinations. In particular, individual destination areas may identify and suggest outdoor programs (e.g., designated paths and parkours tailored for the summer season) to enhance the outdoor experience in terms of comfort and well-being. Future research on the capacity of different forest types in providing regulating services could further promote the planning and use of specific forest areas.

**Author Contributions:** Conceptualization, M.F. and E.G.; methodology, M.F., A.C., F.C. and E.G.; formal analysis, E.G.; investigation, A.C., F.C. and E.G.; data curation, F.C. and A.C.; writing—original draft preparation, E.G, F.C., A.C. and M.F.; writing—review and editing, E.G., F.C, A.C. and M.F.; visualization, E.G. and A.C.; supervision, E.G.; funding acquisition, E.G. All authors have read and agreed to the published version of the manuscript.

**Funding:** This research was realized with the contribution of the Autonomous Province of Trento, Forests and Fauna Service. The APC was funded by the Parco Naturale Paneveggio–Pale di San Martino.

**Institutional Review Board Statement:** Not applicable.

**Informed Consent Statement:** Not applicable.

**Data Availability Statement:** The data presented in this study are available on request from the corresponding author.

**Acknowledgments:** We are grateful to Parco Naturale Paneveggio–Pale di San Martino for the help in field work. We would like to thank the reviewers for the useful and constructive suggestions, which led to a significant improvement in the manuscript.

**Conflicts of Interest:** The authors declare no conflict of interest.

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
