# Peer review of "Forests Attenuate Temperature and Air Pollution Discomfort in Montane Tourist Areas"

_forests, doi:10.3390/f14030545_

Round 1
Reviewer 1 Report (Previous Reviewer 5)
I appreciate the additional work that the authors have done to put the manuscript together. This is an interesting study which has a great potential to offer a contribution. However, the authors need to follow the below comments/ suggested revisions to improve their paper.
1. Hypothesis are still missing.
2. In terms of conclusion and policy implications, it is necessary to clearly add contributions to theory and practice.
Author Response
We would like to thank the reviewer for the appreciation and for the useful and constructive suggestions, which led to improve the manuscript.
- The study hypothesis has been better specified as follows:
“In this study, the hypothesis to be tested is that in Norway spruce-dominated montane forest stands air temperature and air pollution are significantly lower than in neighboring open field sites.”
2. Conclusions paragraph has been modified trying to implement the suggested revisions. Specifically, the following sentences have been added/modified:
“Several studies deal with the capacity of plants to lower air temperatures during the summer by absorbing a fraction of incoming sunlight and by transpiration, and with the theoretic mechanisms of air pollutant removal by plants through both stomatal and non-stomatal processes”.
“This evidence has clear practical implications, especially relevant for the alpine region that may become even more attractive in summer due to the expected rapid increase in air temperature. In general, considering the priorities identified by the EU tourism policies, the role of forests in providing regulating and cultural ecosystem services could be better promoted to favor sustainable, high-quality destinations. In particular, individual destination areas may identify and suggest outdoor programs (e.g., designated paths and parcours tailored for the summer season) to enhance the outdoor experience in terms of comfort and well-being. Future research on the capacity of different forest types in providing regulating services could promote further the planning and use of specific forest areas. “
Reviewer 2 Report (New Reviewer)
Dear author/s,
Really great job with the article. I enjoyed reading it. Only one suggestion - what are theoretical contributions? You can add this in the conclusion section as well as future research and limitations.
This study aimed to demonstrate microclimate regulation and pollutant removal as particularly relevant ecosystem services when considering the tourist appeal of alpine regions. He specifically focused on the Norway spruce mountain forest and answered the question of whether it can effectively reduce air temperature and air pollution. This topic has not been studied enough, so this study is more than necessary and confirms the interest and originality of this research. It fills the literature gap.
The manuscript is well-written, understandable, and easy to understand. It is clear and easy to read even for someone who does not specialize in the subject.
The discussion is very good, only the conclusions lack theoretical contributions as well as future research and limitations. It can improve the conclusion section.
In addition, this study significantly adds novelty and originality to the literature.
Wish you all the best with future work!
Kind regards,
Reviewer
Author Response
We would like to thank the reviewer for the appreciation and for the useful and constructive suggestions, which led to improve the manuscript.
Conclusions paragraph has been modified trying to implement the suggested revisions. Specifically, the following sentences have been added/modified:
“Several studies deal with the capacity of plants to lower air temperatures during the summer by absorbing a fraction of incoming sunlight and by transpiration, and with the theoretic mechanisms of air pollutant removal by plants through both stomatal and non-stomatal processes”.
“This evidence has clear practical implications, especially relevant for the alpine region that may become even more attractive in summer due to the expected rapid increase in air temperature. In general, considering the priorities identified by the EU tourism policies, the role of forests in providing regulating and cultural ecosystem services could be better promoted to favor sustainable, high-quality destinations. In particular, individual destination areas may identify and suggest outdoor programs (e.g., designated paths and parcours tailored for the summer season) to enhance the outdoor experience in terms of comfort and well-being. Future research on the capacity of different forest types in providing regulating services could promote further the planning and use of specific forest areas. “
Reviewer 3 Report (Previous Reviewer 1)
I reviewed this paper as former ID: forests-1850464. After the evaluation, the paper addressed my comments sufficiently, and the research novelty was well improved. It can be accepted after a careful minor revision: the title should be rearranged to prevent misleading.
Author Response
We would like to thank the reviewer for the appreciation and for the useful and constructive suggestions, which led to improve the manuscript.
The title has been changed, as suggested, in:
“Forests attenuate temperature and air pollution discomfort in montane tourist areas.”
This manuscript is a resubmission of an earlier submission. The following is a list of the peer review reports and author responses from that submission.
Round 1
Reviewer 1 Report
The study investigated the attenuation effects of forests on temperature and air pollution discomfort in montane tourist areas with field trips. Although the topic is interesting, relevant studies have been conducted worldwide, similar topic can be seen in BAE, SCS and STOTEN, and the results do not have a sufficient impact to the knowledge base in the current format. The conventional method and the lack of novelty make the quality of the study cannot meet the required standard of the journal. There are some concerns and issues that are related to the clarity of the paper, as:
1. Please rearrange the title as ''The attenuation effects of forests on temperature and air pollution discomfort in montane tourist areas'
2. Novelty unclear: What is the original contribution of the study? The introduction section is not very enlightening on the subject and the reader must read the entire manuscript to see it. Novelty should be made as clear as possible, preferably in relation with other similar studies before the manuscript is accepted for publication.
3. How to measure the NO2 and O3, please add a detailed description.
4. As mentioned in the title, the authors would determine the reduction effects of forests on air temperature and pollution, but after checking the main content, the authors ignore the air pollution, especially in the results section.
5. The limitations of this study should be highlighted.
6. Some results were common and have few contributions to the knowledge base.
Reviewer 2 Report
The main weakness of this paper by my opinion is the too limited and very specific range of circumstances regarding the environmental
parameters : only one spring -summer season - 23May- 07August in 2013 . The summer tourist season in South and Middle
Europe is essentially wider and I think that it need to take its end at least to 15 September. It need also note that May 2013 is very extremely for the space weather conditions - high solar flare activity and follows by them radiation storms and
energetic solar protons penetration in the middle Earth atmosphere (20-90km). Such phenomena affect strongly the O3 and NOx concentrations in the stratosphere over the polar and sub-polar regions in time scales up to few weeks after the corresponding solar events. (see Jackmann et al, 2009 and cites therain ) . Due the atmospheric transport processes (winds and diffusion) these processes could affect over the NOx concentrations on heights in range ~ 1500m or more over the sea level (h) in mountain sites which are on distances even few thousand kilometers with time delay from 1 to 12 -15 months after the corresponding events in the polar regions (Komitov et al., 2016). The typical relative changes over the NOx concentrations is ~20-25% during the period 2006-2013 in Rozhen background small gases monitoring station (Rozhen Observatory, Rodope Mountain, Bulgaria, h=1750m). The meteorological conditions (T, RH etc) from year to year could also varied significantly.
I think that these effects could briefly adds in the section "Discussion" in the course which relates NOx. On the other hand I strongly reccomend to include observation results from at least few years for the same region in a future investigation.
Jackman, C. H., Marsh, D. R., Vitt, F. M., Garcia, R. R., Randall, C. E., Fleming, E. L., and Frith, S. M., 2009, Long-term middle atmospheric influence of very large solar proton events, J. Geophys. Res., 114, D11304, doi:10.1029/2008JD011415
Komitov, B., Dechev, M., Duchlev, P.. Formation of nitrogen oxides
in the Earth's atmosphere by solar proton flares. Bulgarian Astronomical
Journal, 24, 2016,
Reviewer 3 Report
This manuscript significantly explains the positive correlation that forest has on meteorological conditions through actual data and statistical analysis methods. However, some improvements are needed to the manuscript before publication.
First, it seems that the contents of the introduction need to be modified. The purpose of this study seems to be to emphasize the health effects of forest tourism. However, the introduction just focuses on the theoretical ecological services of forests and the EU clean air policy, so the necessity and purpose of the study is not delivered properly. Additional content that can be linked to ecotourism needs to be added at introduction section.
Second, the core index of this study is THI, and it seems that additional discussion on THI is necessary. For example, it would be good if an explanation of the influence of the Mediterranean climate on seasonal changes in RH and temperature and the change in THI according to it would be added.
Finally, check the journal guidelines for manuscript format such as reference citation format or paragraph structure, and edit the manuscript form accordingly.
Reviewer 4 Report
Thank You for providing me an opportunity to read and revise this interesting and detailed paper. I have several sugestions for improving the quality of its content. I suggest to extend discussion and conclusion sections by providing several sections, such as practical implications, limitations and future research proposals. It could also be beneficial to add more explanations in terms of importance of this topic for other sectors, such as tourism. Authors mentioned this sector in the literature review and later in the discussion, however, I believe that authors would improve the quality of the paper by indicating the importance of gained results for other sectors.
Reviewer 5 Report
I appreciate that the paper deals with an interesting issue nowadays and has some noteworthy outputs. However, the authors need to follow the below comments/ suggested revisions to improve their paper.
1. In the introduction it is necessary to specify the purpose and originality of the study in relation to the results identified in the academic literature.
2. I suggest to authors to include in the literature review and discussion section the following:
- Moomaw, W. R., Masino, S. A., & Faison, E. K. (2019). Intact forests in the United States: Proforestation mitigates climate change and serves the greatest good. Frontiers in Forests and Global Change, 27.
- Jennings, L. N., Douglas, J., Treasure, E., & González, G. (2014). Climate change effects in el yunque national forest, puerto rico, and the caribbean region. Gen. Tech. Rep. SRS-GTR-193. Asheville, NC: USDA-Forest Service, Southern Research Station. 47 p., 193, 1-47.
- Scott, D., Jones, B., & Konopek, J. (2007). Implications of climate and environmental change for nature-based tourism in the Canadian Rocky Mountains: A case study of Waterton Lakes National Park. Tourism management, 28(2), 570-579.
- Colloff, M. J., Doherty, M. D., Lavorel, S., Dunlop, M., Wise, R. M., & Prober, S. M. (2016). Adaptation services and pathways for the management of temperate montane forests under transformational climate change. Climatic Change, 138(1), 267-282.
- Muhati, G. L., Olago, D., & Olaka, L. (2018). Land use and land cover changes in a sub-humid Montane forest in an arid setting: A case study of the Marsabit forest reserve in northern Kenya. Global Ecology and Conservation, 16, e00512.
3. Hypothesis are missing.
4. In terms of conclusion and policy implications, it is necessary to clearly add contributions to theory and practice.
Round 2
Reviewer 1 Report
After checking the revised version, I still think the novelty of the revised version cannot meet the standard of Forests journal. Related topics have been studied worldwide, and the results have few new comes for the present knowledge base.
Reviewer 3 Report
The answer to the comment is unclear, so it is not possible to confirm which part has been modified or added. In particular, if the manuscript has been revised, the conclusion part also needs additional revision, but nothing has changed. Also, there is a blank space in lines 146 and 253. Overall, the revision seems to be careless.
Specific indications of corrections are required on the answer sheet. In addition, it is necessary to revise the manuscript by adding the limit part of the thesis.